

# The relationship between social rank and spatial learning in pheasants, *Phasianus colchicus*: cause or consequence?

Ellis J.G. Langley, Jayden O. van Horik, Mark A. Whiteside, Christine E. Beardsworth and Joah R. Madden

Centre for Research in Animal Behaviour, University of Exeter, Exeter, United Kingdom

## ABSTRACT

Individual differences in performances on cognitive tasks have been found to differ according to social rank across multiple species. However, it is not clear whether an individual's cognitive performance is flexible and the result of their current social rank, modulated by social interactions (social state dependent hypothesis), or if it is determined prior to the formation of the social hierarchy and indeed influences an individual's rank (prior attributes hypothesis). We separated these two hypotheses by measuring learning performance of male pheasants, *Phasianus colchicus,* on a spatial discrimination task as chicks and again as adults. We inferred adult male social rank from observing agonistic interactions while housed in captive multi-male multi-female groups. Learning performance of adult males was assayed after social rank had been standardised; by housing single males with two or four females. We predicted that if cognitive abilities determine social rank formation we would observe: consistency between chick and adult performances on the cognitive task and chick performance would predict adult social rank. We found that learning performances were consistent from chicks to adults for task accuracy, but not for speed of learning and chick learning performances were not related to adult social rank. Therefore, we could not support the prior attributes hypothesis of cognitive abilities aiding social rank formation. Instead, we found that individual differences in learning performances of adults were predicted by the number of females a male was housed with; males housed with four females had higher levels of learning performance than males housed with two females; and their most recent recording of captive social rank, even though learning performance was assayed while males were in a standardized, non-competitive environment. This does not support the hypothesis that direct social pressures are causing the inter-individual variation in learning performances that we observe. Instead, our results suggest that there may be carry-over effects of aggressive social interactions on learning performance. Consequently, whether early life spatial learning performances influence social rank is unclear but these performances are modulated by the current social environment and a male's most recent social rank.

Corresponding author
Ellis J.G. Langley,
ejgl201@exeter.ac.uk

## INTRODUCTION

To understand how cognitive abilities may have been shaped by natural selection, it is important to characterise the causes and consequences of individual differences in cognitive performances (*Thornton, Isden & Madden, 2014*; *Thornton & Lukas, 2012*). An individual's position in a social hierarchy is a critical determinant of an individual's fitness (*Von Holst et al., 1999*) and is likely to be closely linked to their cognitive performance. Social rank greatly influences their access to resources (*Popp & DeVore, 1979*; *Wilson, 1975*), stress (*Abbott et al., 2003*; *Creel, 2001*; *Sapolsky, 2005*) and opportunities for learning (*Chalmeau & Gallo, 1993*). However, it is not clear whether social rank arises as a consequence of pre-existing individual differences in cognitive ability, or if social rank and its associated fitness benefits are a cause of individual differences in cognitive abilities.

Social rank may be predetermined by individual differences in characteristics that influence social success, as described by the *Prior attributes hypothesis* (*Chase et al., 2002*). Cognitive ability may be one such characteristic, with cognitively able individuals going on to achieve dominance (*Humphrey, 1976*; *Byrne & Whiten, 1988*; *Seyfarth & Cheney, 2002*). For example, behavioural inhibition may enable individuals to respond appropriately to competitors (*Amici, Aureli & Call, 2008*) and avoid unnecessary aggression (*Strayer, 1976*). Social learning can inform individuals about conspecifics motivations (*Seyfarth & Cheney, 2002*), as well as their fighting ability and consequently guide future social interactions (fighting fish, *Betta splendens, Oliveira, McGregor & Latruffe, 1998*). Thus, individuals with more proficient social learning abilities have been found to be higher ranking (domestic chickens, *Gallus gallus, Nicol & Pope, 1999*). Similarly, we may expect that general learning ability is associated with high social rank (social success, *Humphrey, 1976*). Learning allows individuals to adapt to changing (social) environments. Performances on operant foraging (starlings, *Sturnus vulgaris,* (*Boogert, Reader & Laland, 2006*) and spatial learning tasks are reported as superior in dominant individuals (pheasants, *Phasianus colchicus, Langley et al., 2018a*; mountain chickadees, *Poecile gambeli, Pravosudov, Mendoza & Clayton, 2003*; mice, *Fitchett et al., 2005*; *Francia et al., 2006*). This may be because individuals that are inherently good at learning are more efficient at beneficial behaviours such as foraging (bumblebees, *Bombus terrestris, Raine & Chittka, 2008*), mate choice (*Dukas & Ratcliffe, 2009*), and navigating the social environment which brings fitness benefits. However, evidence that these differences in performance existed prior to the establishment of dominance is lacking (*Chichinadze et al., 2014*). There has not been an explicit test of whether individual differences in cognitive performance determine social rank.

Alternatively, social rank may be a cause of variation in cognitive performances due to the associated demands of living in a social hierarchy; we term this the *social-state dependent* hypothesis. This may occur via stress (*Abbott et al., 2003*; *Creel, 2001*; *Sapolsky, 2005*), that arises from the immediate social environment and is influential in shaping the expression of individuals' cognitive ability (*De Kloet, Oitzl & Joels, 1999*; *Mendl, 1999*). First, stress may be caused by social pressures and in some cases, the dominant individuals may suffer from high stress and consequently exhibit poorer cognitive performances. When crab-eating macaques, *Macaca fascicularis,* were placed into different social groups, a natural decrease

in rank was accompanied by a decrease in errors on object and colour discrimination and reversal tasks (*Bunnell & Perkins, 1980*; *Bunnell, Gore & Perkins, 1980*). The authors suggest that the differences in performance between high and low ranking macaques was due to the chronic social stresses experienced by dominant individuals when maintaining their social rank (*Bunnell, Gore & Perkins, 1980*). Dominant crabs, *Chasmagnathus granulatus,* demonstrated shorter memory retention of a dangerous signal (context-signal-memory), but only after a dominance encounter, and not before (*Kaczer, Pedetta & Maldonado, 2007*), suggesting that the aggressive encounter was detrimental to the aggressor.

In some cases, the subordinate individuals may exhibit poorer cognitive performances due to aggression received. The acquisition of dominance status affected spatial learning ability in mice, *Mus musculus* (*Barnard & Luo, 2002*), with the individual of a dyad that became subordinate exhibiting impaired performance. The authors suggest that this difference was mediated through aggression as there was a negative relationship between learning performance and the number of aggressive acts received after paired housing. Impairment in subordinate's spatial learning ability also persist in mice, even after previously paired individuals were isolated and social pressures of rank had been removed (*Fitchett et al., 2005*).

A second source of stress, resulting from social rank is that of nutritional stress caused by the unequal distribution of resources across a social hierarchy (*Wilson, 1975*; *Popp & DeVore, 1979*). Dominants are often larger than subordinates (red-deer, *Cervus elaphus, Clutton-Brock, Guinness & Albon, 1982*; carrion crows, *Corvus corone corone*, *Richner, 1989*; red-winged blackbirds, *Agelaius phoenix, Searcy, 1979*) and in many cases more aggressive (*Chase et al., 2002*), thus winning access to resources. Alternatively, in some species, social rank is maternally inherited and hence relatedness determines access to resources (Japanese macaques, *macaca fuscata, Kawamura, 1958*). Dominant individuals are reported to be in better body condition than subordinates (great tits, *Parus major, Carrascal et al., 1998*; red-deer, *Clutton-Brock, Albon & Guinness, 1984*). Improved nutrition may decrease stress overall and additionally dominant individuals may have more energy to invest in costly cognitive abilities (*Aiello & Wheeler, 2009*).

Social rank may influence opportunities for learning and affect cognitive performances. Subordinate chimpanzees, *Pan paniscus,* were unlikely to interact with a cognitive task when the dominant individual was present (*Chalmeau & Gallo, 1993*). In addition to affecting opportunity, social rank may affect the voluntary expression of cognitive ability. Subordinate rhesus macaques, *Macaca mulatta,* that had previously solved a food choice task, did not express these behaviours in the presence of dominant individuals (*Drea & Wallen, 1999*). Consequently, the differences between the social ranks in stress (social and nutritional) and opportunity may each contribute to social rank-related variation in cognitive performances.

The pheasant, *Phasianus colchicus,* offers a suitable system in which to explore causality in the relationship between cognitive performances and social rank. Pheasants are a precocial species and large numbers can be hatched on the same day and reared without parents. Pheasant chicks can be assayed for cognitive performance using batteries of psychometric tests under captive conditions (*Van Horik et al., 2017*) prior to their release into the wild. Once in the wild, pheasants exhibit harem defense polygyny and males engage

in agonistic interactions (*Hill & Robertson, 1988*). Winners of these interactions are more likely to become dominant territory holders and attract females. Losers of these interactions become satellite males who do not hold fixed territories and are subordinate to territory holders and likely obtain low reproductive success. Territory acquisition begins as early as October (*Ridley & Hill, 1987*; *Whiteside et al., 2018*) and territory holders have smaller, more concentrated home ranges than subordinate satellite males (*Grahn, Goransson & Von Schantz, 1993*). Male pheasants exhibit behavioural indicators of dominance, such as crowing (*Ridley & Hill, 1987*; *Heinz & Gysel, 1970*) and lateral displays (*Hill & Robertson, 1988*), and captive studies demonstrate that dominant males perform these dominance display behaviours at a significantly higher rate than subordinates (*Mateos & Carranza, 1999*). These displays are suggested to attract females (*Mateos & Carranza, 1999*) and deter competitors (*Hill & Robertson, 1988*; *Ridley & Hill, 1987*). In captivity, when males are housed in groups they establish stable hierarchies over short periods at least (*Mateos & Carranza, 1997a*; *Mateos & Carranza, 1997b*), and the higher ranking males have preferential access to females and dominate particular areas of the housing aviary (E Langley, pers. obs., 2015). Winners of dyadic interactions in the field match those in captivity, thus, male dominance in captivity reflects the situation in the wild (*Von Schantz et al., 1989*). We have previously shown that variation in performance on a spatial discrimination task is associated with social rank in adult male pheasants, which were tested while housed in a group with an established social hierarchy (*Langley et al., 2018a*). Perhaps, male pheasants that are inherently good at learning about space become dominant because they are better able to recall spatial features and so more efficiently establish and hold a territory. Alternatively, dominant males with smaller home ranges may express better spatial learning performances because they have had more opportunity to learn spatial cues in a reliable and consistent territory (i.e., they learn to learn).

We investigated whether the ability to discriminate between locations was a pre-requisite to male pheasants' social rank, or whether this ability is more likely a consequence of social rank. We assayed the cognitive performance of pheasant chicks before we released them into the wild. Then, prior to the breeding season that begins in March and lasts until May (*Göransson et al., 1990*), we captured adults from the wild. Individuals are captured at this time so that their eggs can be collected for incubation, as part of a larger experiment. We expect that measures of social rank are more meaningful during these months because this is when males are in intense competition for resources, i.e., access to females. We assessed adult males' group social rank while housed in a multi-male multi-female group aviary and also manipulated dominance rank by housing males singly, in a non-competitive, multi-female condition, which we term the 'perceived dominance' condition. Hence, in this condition, males were provided with an uncontested territory, a harem of females and no direct social pressure from other males. While males were in this perceived dominance condition and experiencing equivalent social ranks, we assayed their performance on the same task that we had presented to the chicks. To test whether a male's cognitive performance may be the cause of, or a consequence of social rank, we asked three questions. First, is an individual's cognitive performance consistent from chick to adult? For a cognitive ability to be a prior determinant of social rank, we expected individual cognitive performances to be consistent

from chick to adult, as this would indicate cognitive ability developed outside of and prior to dominance interactions. If cognitive performances are not consistent from chick to adult this suggests that they may be altered in response to an individual's current social environment. Second, we asked whether chick cognitive performances predict their future social rank, suggesting that a prior ability in this domain may determine subsequent social rank. Positive results for question 1 and 2 would provide support for the prior attributes hypothesis. Third, we tested adult males' cognitive performances while they were housed in the perceived dominance condition and investigated whether this was related to their captive social rank. Critically, we assessed whether this perceived dominance condition was associated with increases in "*dominance-display*" behaviours; crowing and lateral struts, as an indication of the effectiveness of the rank manipulation. If inter-individual variation in cognitive performance while experiencing this rank manipulation is not explained by an individuals' most recent social rank, this provides support for the social state dependent hypothesis; because all males were experiencing the same social rank and therefore performance on the task is expected to be similar among males.

## METHODS

### Study system, subjects & housing

This study was conducted from May 2015–June 2016 at North Wyke Rothamsted Research Farm, Devon (50°77′N, 3°9′W). We reared 194 pheasant chicks from hatching in one of four identical aviaries. Chicks were identifiable by numbered patagial wing tags (Roxan Ltd). For the first two weeks of life, chicks had access to an indoor 2 m × 2 m heated aviary. At three weeks they also had access to a covered but unheated 1 m × 4 m outdoor run and at four weeks they also had access to a 4 m × 12 m outdoor aviary. Throughout the aviaries, chicks had access to perches and food and water *ad libitum*. Within the indoor section of the aviary, chicks could enter a testing chamber through a sliding door and engage in cognitive testing and exit to the outdoor area via a lift-up door. One hundred and forty-nine chicks participated in the task described in this study. When the chicks were 10 weeks old, we released them on to the site that covers 250 Ha of which there is lowland deciduous woodland, grassland, fen meadow and 40 artificial wheat feeders.

In March 2016 we caught adult pheasants (≥10 months old) using baited funnel traps. The catching period lasted for three weeks, by the end of which we had caught most of the males on the site, as determined by field observations. We caught 22 males, 11 of which we had reared as chicks, hereby referred to as *known* males and the remaining 11 males were of unknown rearing history, hereby referred to as *unknown* males. Known males that we did not catch either died or dispersed off of the site. Adult males were assigned to one of two different outdoor aviaries/social conditions, either; a large group aviary (19 m × 23 m), containing multiple females to give a male to female ratio of 60:40; or assigned to one of 10 smaller identical aviaries (4 m × 8 m), in which males were housed individually with either two or four females. The allocation of two or four females was determined at random and formed part of a separate experiment on female cognitive performance (*Langley et al., 2018b*). Aviaries were in visual but not auditory isolation from each other. All aviaries

contained elevated perches, refuge areas, and food and water *ad libitum.* Males experienced both social conditions and a general overview of the methods is shown in Fig. 1.

## Cognitive test apparatus

Spatial learning and memory tasks on avian subjects typically investigate subjects' ability to reliably locate a food reward on a foraging apparatus containing wells (Western scrub jays, *Aphelocoma californica, Pravosudov, Lavenex & Omanska, 2005*; zebra finch, *Taeniopygia guttata, Sanford & Clayton, 2008*; song sparrows, *Melospiza melodia, Sewall et al., 2013*; New Zealand North Island robins, Petroica longpipes, *Shaw et al., 2015*). Wells may be concealed by flaps (*Sanford & Clayton, 2008*) or filled with sand (*Pravosudov, Lavenex & Omanska, 2005*), thus requiring the subject to search the locations to retrieve food rewards. Our 'top-bottom' discrimination task required subjects to discriminate between two identical wells arranged vertically on a rectangular apparatus (38 cm × 14 cm × 4 cm). The top well, furthest from the bird, contained a mealworm food reward. The bottom well, closest to the bird, was unrewarded and blocked by a bung. Both wells were covered with a layer of opaque crepe paper which chicks and adults were trained to peck through prior to testing. Both wells were unmarked and identical and were only distinguishable by their location on the task apparatus (top vs. bottom). During a trial we allowed individuals to make one choice per pair of wells. If individuals chose correctly, indicated by pecking at the crepe paper of the rewarded well, we allowed the individual to consume the food reward before the wells were removed. If individuals chose incorrectly, indicated by pecking at the crepe paper of the unrewarded well, the wells were removed and a new pair of wells was presented.

## Chick training and cognitive testing

From one day old chicks were habituated to human experimenters. We trained chicks to enter a testing arena in groups and allowed them to become familiar with the testing apparatus by placing mealworms in open wells and on top of the apparatus so that they were visible to the chicks. In subsequent training sessions we presented groups of chicks with mealworms only within the wells to encourage individuals to search for rewards within the wells. Following this, we added broken crepe paper onto the wells and over multiple sessions the wells became increasingly concealed until individuals spontaneously pecked through the crepe paper. At approximately three weeks old, chicks were trained to individually enter the testing chamber, located behind a sliding door, upon hearing an auditory command (whistling/humming from a human experimenter). Individuals could voluntarily enter the testing chamber during a training or testing session and the order in which they enter is consistent (*Van Horik et al., 2017*). Cognitive testing began when individuals could competently peck through the crepe paper to retrieve the mealworm reward. Each testing session consisted of 10 trials. Once the trials were completed, chicks were released through the exit door. If a chick displayed signs of stress, such as flapping or lost calling, or they did not interact with the apparatus after two minutes, they were released through the exit door. We tested individuals at eight weeks old for three sessions over two days. There were two morning sessions on consecutive days, beginning at 9 am

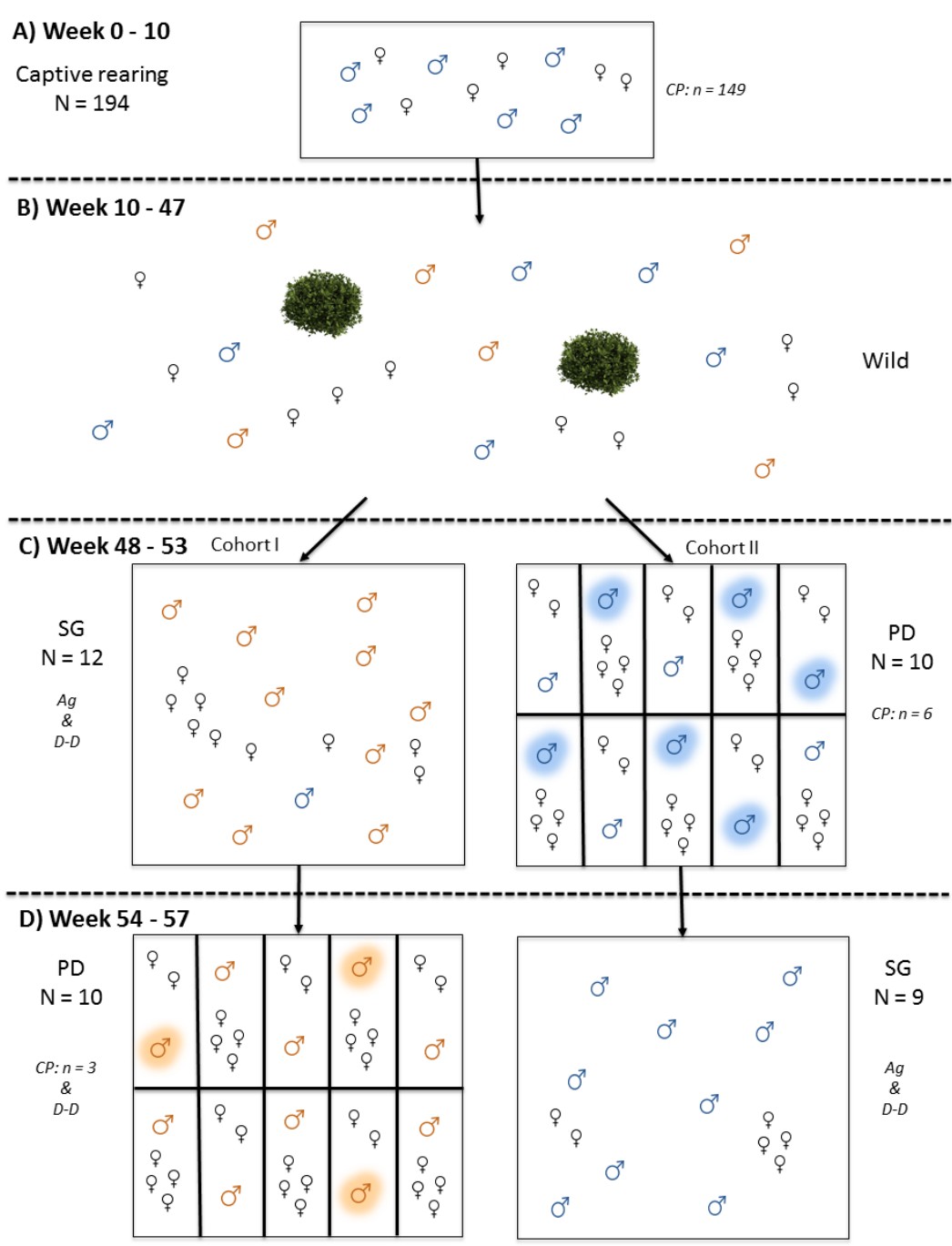

**Figure 1 Timeline of experimental procedures.** (A) Chick rearing, (B) Release into the wild, (C) Adult housing; Cohort I in the Social Group (SG) condition and Cohort II in the Perceived Dominance (PD) condition, (D) Adult housing; cohorts switched conditions. Blue birds, known; orange birds, unknown; *N*, total sample size in each condition; CP, cognitive performance assayed; D-D, dominance-display behaviours recorded; Ag, agonistic interactions between males recorded; *n*, sample size of those tested on the spatial discrimination task (these individuals are also highlighted on the figure).

**Table 1** Ethogram of agonistic interactions between male pheasants collected during the SG condition for the inference of social rank.

| | |
|---|---|
| *Agonistic* | |
| *Chase* | aggressor (winner) runs towards opponent and opponent flees (loser) |
| *Threat* | aggressor (winner) steps forwards and makes a sharp lunge towards opponent (loser), generally, the opponent flees. Similar to the start of a *chase* but aggressor does not continue to run |
| *Contact* | aggressor (winner) pecks opponent (loser) with the bill, usually directed at the head, or aggressor (winner) jumps at opponent feet first to direct spurs at opponent (loser) |
| *Submissive* | |
| *Avoid* | an individual (loser) rapidly changes trajectory while walking and is within 3 m of another individual (winner) that is not showing any apparent signs of aggression |

and lasting until approximately 11:30 am once all chicks had been tested. Between these two sessions, chicks received one afternoon session beginning at 14:00 pm and lasting until approximately 16:30 pm. Food was removed from the aviaries one hour prior to testing. The three testing sessions resulted in a maximum of 30 trials per individual. Olfactory cues were not controlled for but because galliformes have a poor sense of smell (*Corfield et al., 2015*), we were confident that individuals were not using olfactory cues to locate the rewarded well.

## Adult social conditions

Eleven unknown males and one known male (chosen at random) that we captured as adults were assigned to the 'Social Group (SG)' condition (the large group aviary). The remaining ten known males were assigned to the 'Perceived Dominance (PD)' condition (one of 10 individual aviaries). We housed known males in the same social condition so that we could compare their cognitive performances to their social rank, relative to the other males that they were reared with as a chick. Hence, we did not assign males to the conditions in a randomised way. Due to low participation on cognitive testing from known males while in the PD condition (see below), we also placed unknown males in to the PD condition afterwards to assay their learning performance in an attempt to increase our sample size. Hence, males experienced both conditions; those assigned to the SG condition first and then the PD condition are hereby referred to as 'cohort I', and those experiencing the conditions in reverse, are referred to as 'cohort II'.

### Social Group (SG) condition

We collected observations *ad libitum* on the outcomes of dyadic agonistic interactions between males for the inference of social rank (Table 1) and dominance-display behaviours as an indicator of perceived social rank (Table 2). There were two observers at a given time each monitoring different areas of the aviary to ensure all behaviours were recorded. Observers were visually concealed from the birds. For the recording of dyadic agonistic interactions we assigned a winner and a loser. For the recording of dominance-display

**Table 2** Ethogram of display behaviours characteristic of dominance collected while pheasants were housed in both the SG and the PD conditions.

| | |
|---|---|
| *Lateral strut* | Male lowers head and flattens one wing toward receiver, primaries may touch the ground while erecting ear tufts and inflating wattle. Tail is spread. Sometimes the display is accompanied with vibration of the tail to create audible sound. |
| *Crow* | Loud, sudden two-syllable call. Followed by a brief and loud wing flap (*Heinz & Gysel, 1970*; *Cramp & Simmons, 1980*) |

behaviours, we calculated rate (event/hour) of each behaviour performed by each male. For cohort I males, observations were collected during weeks 48–53 on 12 males, prior to cognitive testing in the PD condition (Fig. 1). For cohort II males, observations were collected during weeks 54–57 on 9 males (one male died unexpectedly before being placed in the SG condition), after cognitive testing in the PD condition (Fig. 1).

### Perceived Dominance (PD) condition

Housing in one of these 10 aviaries provided the male with an exclusive territory, a harem and no direct social pressure from other males. Males were randomly allocated to an aviary containing two or four females. This simulated the male holding a high social rank. Males had their cognitive performance assayed while in this condition only. Outside of cognitive testing we also collected behavioural observations on dominance-display behaviours (Table 2) as an indication of a male's perceived social rank for cohort I males. Five of the aviaries could be observed simultaneously and the dominance behaviours were conspicuous. Each day we determined randomly which five aviaries to observe for the first 30 min and then observed the remaining five aviaries for 30 min. Observations begun at variable times of the day to account for differences between males in their activity levels. For each individual, we calculated rate per hour of each of the two dominance behaviours using the same methods as those used in the SG condition. We did not collect observations on dominance-display behaviours of cohort II males due to time constraints.

### Adult training and cognitive testing procedures

Performances on the spatial discrimination task were assayed while males were housed in the PD condition. We habituated all individuals to approach the test apparatus, located in the testing area of their aviary (Fig. 2). The apparatus was located between two opaque screens so that it could only be approached and viewed by a bird 'front-on'. These screens were necessary for the testing of adults because the females within the pen were also tested on this task, as part of a separate experiment (*Langley et al., 2018b*); we wanted to prevent social learning of task affordances and these screens allowed only the bird being tested to view the apparatus. To signal to the males that the apparatus was available, a visual cue (black and white swirl pattern) was placed on the wall in the testing area and we tapped and scratched the apparatus, which was situated in the corner of the aviary. We used similar methods to the chick training regime by heavily baiting the box with mealworms so that they were visible, with the gradual transition to only placing worms within the wells and the addition of crepe paper covering the wells. We attempted to train all 20 males while

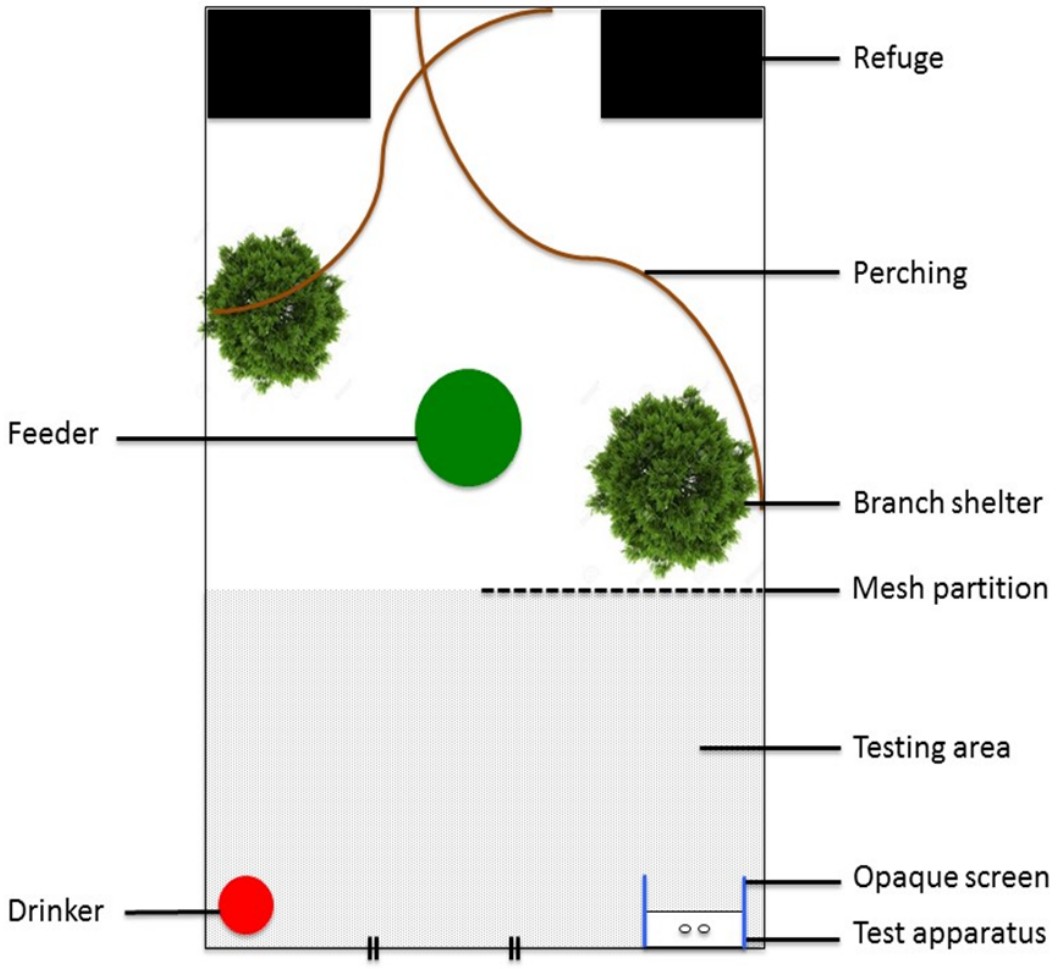

**Figure 2  Aerial view of single housing pen of the Perceived Dominance (PD) condition with testing area and test apparatus.**

they were housed in the PD condition, but the males proved difficult to test and appeared distracted by females during the breeding season. We ceased in our attempts to train males that did not interact with the task apparatus on five consecutive training sessions. Three unknown males of cohort I and six known individuals of cohort II reliably participated in the task. During most test sessions, we were required to use a temporary mesh partition that stopped the females from approaching males while they were interacting with the test apparatus. The use of this mesh partition did not appear to be stressful as males readily engaged in cognitive testing shortly after the partition was implemented. Males were not caught or handled during testing. Each testing session consisted of 20 trials and begun between 8 am to 5 pm. We chose the order in which to test males at random. If a male did not engage with the task within 10 min, we moved on to another male and tried the initial male later that day and did not repeat the same order of testing on any other day. Individuals received one session per day, for five days, resulting in 100 trials in total. We

suspected that this high number of trials would provide us with more detailed learning curves and increase our ability to detect differences between males of different social rank in the rate of learning. We did not include the order of testing in analyses due to small sample size and this variable not being of interest. However, because the task was voluntary, we suspect that once males begun the task, they were equally motivated to engage in testing. Due to adverse weather conditions, cognitive testing for Cohort II was delayed and we ran out of time to replicate this duration for Cohort I; Cohort I and II were housed in the PD condition for 11 and 23 days, respectively, before cognitive testing begun.

## Statistical analysis

All analyses were conducted using R v.3.3.3 (*R Core Team, 2017*).

### Social rank

For each cohort we inferred the social hierarchy using the same methods as those in *Langley et al. (2018a)* using the winner-loser data of agonistic and submissive interactions (Table 1), we generated 'Randomized Elo-ratings' using the *aniDom* package (*Farine & Sanchez-Tojar, 2017*) and assessed hierarchy uncertainty using the two methods described in *Sánchez-Tójar, Schroeder & Farine (2017)*. First, we estimated repeatability of the individual Elo-ratings generated from replicated datasets ($n = 1{,}000$) using the *rptR* package (*Schielzeth, Stoffel & Nakagawa, 2017*), with high repeatability scores indicating a steep hierarchy (high probability that a dominant individual wins a contest). Second, we split the interaction dataset into two halves, computed 1,000 individual ranks for each half using the randomized Elo-rating method and calculated the Spearman's Rank Correlation $r_S$ between the ratings generated by the two halves. We report the mean $r_S$ and 95% confidence interval range of the correlation values. These results indicated high levels of certainty in the data, therefore we used the mean of the randomized Elo-ratings from the full dataset in subsequent analyses, hereby referred to as 'mean Elo-rating'.

### Cognitive performance

We generated learning curves using a binary logistic regression model (GLM) for each individual that performed the top-bottom discrimination task as a chick and as an adult ($n = 6$), using the first 20 trials for both chicks and adults so that learning curves were comparable. From these curves we calculated the probability that an individual would choose correctly on their final trial ($X = $ Final), which is derived from solving the equation $Y = 1/(1 + \exp[-(b_0 + b_1 X)])$, whereby $b_0$ depicts the intercept and $b_1$ depicts the slope estimate from the learning curve GLM. We consider this measure indicative of how well an individual has learned the task by the end of the testing. We also calculated the predicted trial number when an individual reaches or will reach a learning criterion of 80% probability of choosing correctly ($Y = 80$), this is derived by solving the equation $X = (-\ln 0.25 - b_0)/b_1$. We consider this indicative of how much experience an individual requires to adequately learn the affordances of the task. The $X = $ Final and $Y = 80$ measures were calculated for both chick and adult task performances. We asked three questions to distinguish between directionality in the relationship between cognitive performances and social rank. (1) We tested whether individual learning performances were consistent from

chick to adult, using the *ICC* package (*Wolak, Fairbairn & Paulsen, 2012*) to assess the intra-class correlation between chick and adult $X =$ Final; and chick and adult $Y = 80$. This was conducted on six individuals that completed the task at both ages. (2) We tested whether chick learning performance predicted adult social rank using a Spearman's Rank Correlation between chick $X =$ Final and their adult mean Elo-rating; and chick $Y = 80$ and their adult mean Elo-rating. This was conducted on one individual of cohort I and seven individuals of cohort II, that completed the cognitive task as a chick (two additional individuals to those in question 1; that did not complete adult cognitive testing). (3) Finally, we fit a generalized linear mixed model (GLMM) with a binomial error structure and a logit link function to assess whether adult learning performance (correct: 1 yes / 0 no) was predicted by group social rank (mean Elo-rating). We also included cohort, the number of females that males were housed with during the PD condition, choice on first trial (correct: 1 yes / 0 no) and trial number (2–100) as explanatory variables. A two-way interaction between mean Elo-rating and trial number was included to examine whether individuals differ in their rate of learning in relation to their group social rank. We define rate of learning as the speed at which individuals switch from making a series of incorrect choices to a series of correct choices and is deduced from the steepness of the learning slope (trial*social rank; $b_1$). A main effect of trial is indicative that there was an increase in the probability that males would choose correctly as trial number increased. A main effect of social rank on learning performance is indicative that social ranks differ in their overall accuracy of task performance, inclusive of performance on all trials. We included cohort to account for the order in which males experienced the social conditions, as well as their rearing history (i.e., whether they had experienced this task as a chick). We included the number of females in the PD condition because we have previously shown that group size affects female learning performance (*Langley et al., 2018b*). We included choice on first trial (correct: 1 yes / 0 no) to control for random choice on this first trial; as this trial was prior to the opportunity for learning but may affect subsequent performance on the task and this left the trial variable with trial number 2–100 (after trial 1 was removed). To facilitate convergence we converted trial number and mean Elo-ratings to $z$-scores (Gelman & Hill, 2007). Individual was included as a random term (random intercepts, fixed slopes model). We assessed the fit of this model by comparing it to an equivalent random intercepts and random slopes model and found that the random intercepts only model was adequate ($X^2 = 0.261$, $p = 0.878$) and therefore used this for subsequent analyses. We tested the significance of explanatory variables using likelihood ratio tests. This model was fitted on eight adult males that each completed 100 trials; three males of cohort I and five males of cohort II.

### Dominance display behaviours

We investigated the effectiveness of our rank manipulation and compared rates of dominance-display behaviours of males when they were in the SG condition with the rates of dominance-display behaviour when they were housed in the PD condition, using a Wilcoxon signed-ranks test on all 10 individuals of cohort I. We adjusted the rate of lateral displays directed towards females by controlling for female density by dividing the

mean rate of displays performed by the number of females housed in the aviary. We only included lateral struts which were directed towards females so this was consistent between social conditions (females are present in both social conditions, whereas multiple males were only present in the SG condition and we wanted to avoid introducing bias into our results); lateral displays that were clearly directed towards males or in cases where the receiver was ambiguous, were not included in analyses.

### Ethical considerations

Chicks and adults were habituated to human observation and were subject to minimal handling. All training procedures were adopted to mitigate stress during cognitive testing and birds could choose whether or not to participate in tasks. Experimenters were concealed from view of the birds and birds were reared at a lower density than that recommended by DEFRA's code of practice (*DEFRA, 2009*), thus reducing stress. During capture of adults from the wild, traps were checked at least three times a day. Adult birds were held in captivity for three months, after which they were released at the capture site. All work was approved by the University of Exeter Psychology Ethics Committee and the work was conducted under Home Office licence number PPL 30/3204 to JRM.

## RESULTS

### i. Is learning performance consistent from chick to adult?

Consistency in chick and adult learning performances after 20 trials on the spatial discrimination task was significantly different from zero (Intra-class correlation coefficient = 0.464; 95% confidence interval: 0.373–0.900; Fig. 3). Conversely, the consistency in chick and adult learning performances in the predicted trial number of having reached a learning criterion of 80% was not significantly different from zero (intra-class correlation coefficient = −0.244; 95% confidence interval: −0.816–0.618; Fig. 4). This suggests that individuals are consistent in their accuracy after 20 trials but differ in their predicted number of trials taken to reach a learning criterion.

### ii. Does a chick's learning performance predict adult social rank?

The two uncertainty measures we obtained from using the Randomized Elo-rating method to generate hierarchies for both cohorts, indicate these inferred hierarchies were highly reliable indicators of social rank (Table 3). We found no relationship between mean Elo-rating and chicks' predicted performances at the end of the spatial discrimination task (Spearman's Rank Correlation: $X =$ Final, $r_s = 0$, $n = 8$, $p = 0.99$). Similarly, there was no relationship between the trial number in which chicks were predicted to reach an 80% probability of choosing correctly and their adult social rank (Spearman's Rank Correlation: $Y = 80$, $r_s = −0.524$, $n = 8$, $p = 0.197$).

### iii. Does variation in cognitive performances persist once social rank is standardised?

*Evidence that the social rank manipulation was effective*

Both indicators of dominance (crowing and lateral displays) were expressed at higher rates by males in single male groups than when housed in social groups. Of the ten males of

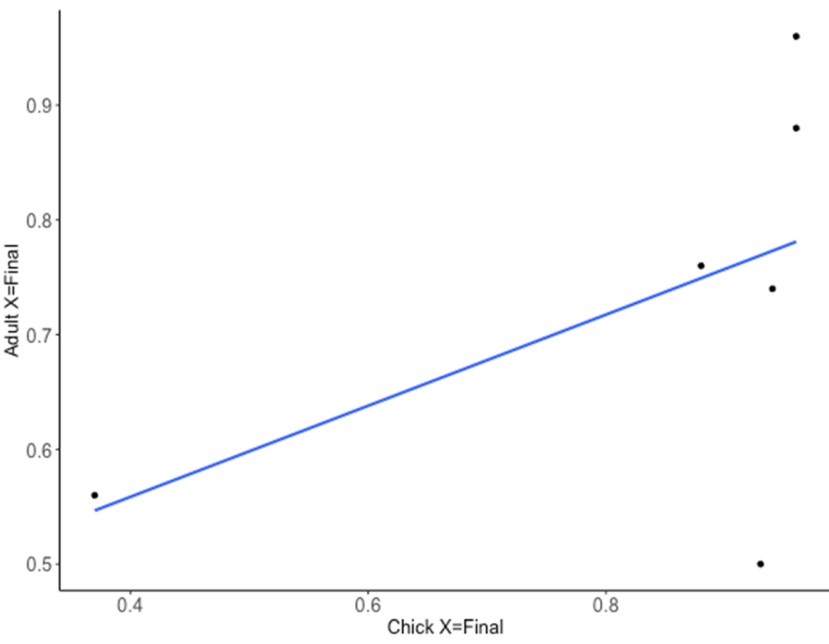

**Figure 3 Relationship between the predicted probability of a correct choice on the final trial (X = Final) for chick and adult spatial discrimination performances (Cohort II, $n = 6$).**

cohort I, two males never crowed in either social condition and eight males increased their rate of crowing behaviour while housed in the rank manipulation (PD condition) compared with when they were housed in the social group (SG) condition (Wilcoxon Signed Ranks test: $n = 10$, $p = 0.014$, Table 4, Fig. 5).

Three of ten males performed fewer lateral struts while in the PD condition compared with when they were housed in the SG condition, whereas seven males performed struts at a higher rate while housed in the PD condition (Wilcoxon Signed Ranks test: $n = 10$, $p = 0.002$, Table 4, Fig. 6).

*Cognitive performance during a rank manipulation*

While adult males were housed in the perceived dominance condition (PD), individuals that had a high social rank when in the social group (SG) condition, learned the spatial discriminations at a faster rate than those of lower social rank (GLMM: Trial number*mean Elo-rating, $X^2 = 12.143$, $df = 1$, $p < 0.001$, Table 5, Fig. 7). The number of females a male was housed with during the PD condition was a significant predictor of spatial discrimination task performances (GLMM: number of females, $X^2 = 11.255$, $df = 1$, $p < 0.001$), with males housed with four females having a higher probability of choosing correctly than males housed with two females (Table 5). High and low ranking males were equally as likely to be housed with four females (Table 6). Whether males were known or unknown did not relate to learning performances and this variable also controlled for the order in which the males experienced the two different social conditions (GLMM: cohort, $X^2 = 0.554$, $df = 1$, $p = 0.456$). Whether a male chose correctly on their first trial did not relate to performance on the remainder of the task (GLMM: first choice, $X^2 = 1.187$,

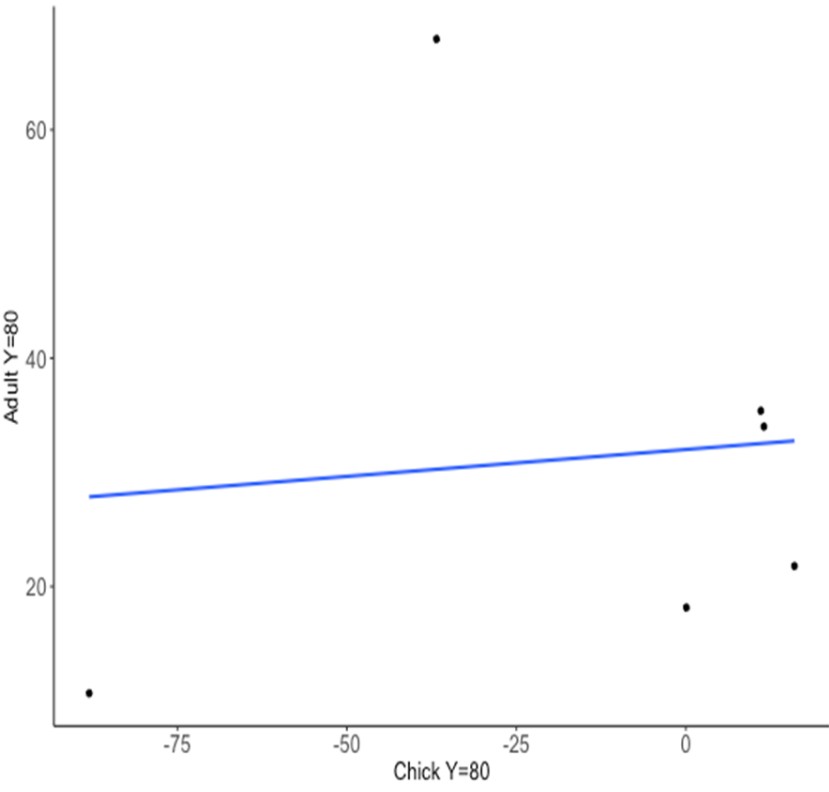

**Figure 4** Relationship between predicted trial number when reached a learning criterion of 80% probability of a correct choice ($Y = 80$) for chick and adult spatial discrimination performances (Cohort II, $n = 6$).

**Table 3 Hierarchy statistics for male pheasants of cohort I and II while housed in the social group condition (SG).**

| $n$ | Cohort | No. interactions | Obs (days) | $r$ | $r^2$ Mean 2.5% 97.5% |
|---|---|---|---|---|---|
| 12 | I | 1,044 | 47 | 0.984 | 0.948 0.881 0.993 |
| 9 | II | 701 | 14 | 0.996 | 0.976 0.917 1.000 |

**Notes.**

$r$, repeatability estimate for individual Elo-ratings generated from replicated datasets; $r^2$, correlation coefficient from Spearman's Rank Correlation between two halves of split dataset.

$df = 1$, $p$ 0.276) and males across the hierarchy were equally as likely to choose correctly or incorrectly on their first choice (Table 6).

## DISCUSSION

The relationship between cognitive performance and social rank is often reported but the issue of whether performance on a cognitive task is a cause or consequence of social rank is seldom considered. We show that cognitive performances on a spatial discrimination task by male pheasants were partly consistent across an individual's lifetime but that chicks' performances failed to predict their adult social rank. Therefore, we

**Table 4  The rate per hour of dominance-display behaviour for male pheasants while housed in each of the two social conditions, in relation to social rank while in the social group condition and the number of females housed with when in the perceived dominance condition.**

| Male | Mean Elo-rating | Females | Lateral strut | | | Crow | | |
|------|------|---------|------|------|------|------|------|------|
| | | | SG | PD | Increase | SG | PD | Increase |
| 1 | 773.8359 | 2 | 0.138 | 0.1 | −0.038 | 6.655 | 11.333 | 4.679 |
| 2 | 403.1283 | 4 | 0.01 | 0.017 | 0.007 | 0.036 | 4.3 | 4.263 |
| 3 | 385.0008 | 4 | 0.096 | 0.167 | 0.071 | 0.145 | 0.333 | 0.188 |
| 4 | 207.2864 | 2 | 0.013 | 0.017 | 0.003 | 0 | 3.467 | 3.367 |
| 5 | −16.0075 | 2 | 0.019 | 0.017 | −0.002 | 0 | 5.4 | 5.400 |
| 6 | −140.793 | 4 | 0.025 | 0.225 | 0.2 | 0 | 0 | 0 |
| 7 | −302.046 | 2 | 0.006 | 0.133 | 0.128 | 0 | 10.767 | 10.767 |
| 8 | −421.245 | 2 | 0.057 | 0.017 | −0.041 | 0 | 0 | 0 |
| 9 | −601.546 | 4 | 0.002 | 0.15 | 0.148 | 0 | 3.233 | 3.233 |
| 10 | −891.792 | 4 | 0.01 | 0.05 | 0.04 | 0 | 0.067 | 0.067 |

**Notes.**

SG, Social Group condition; PD, Perceived Dominance condition; Increase, (PD rate –SG rate).

Lateral strut rate adjusted for female density = 19 females in the SG condition and two or four in the PD condition.

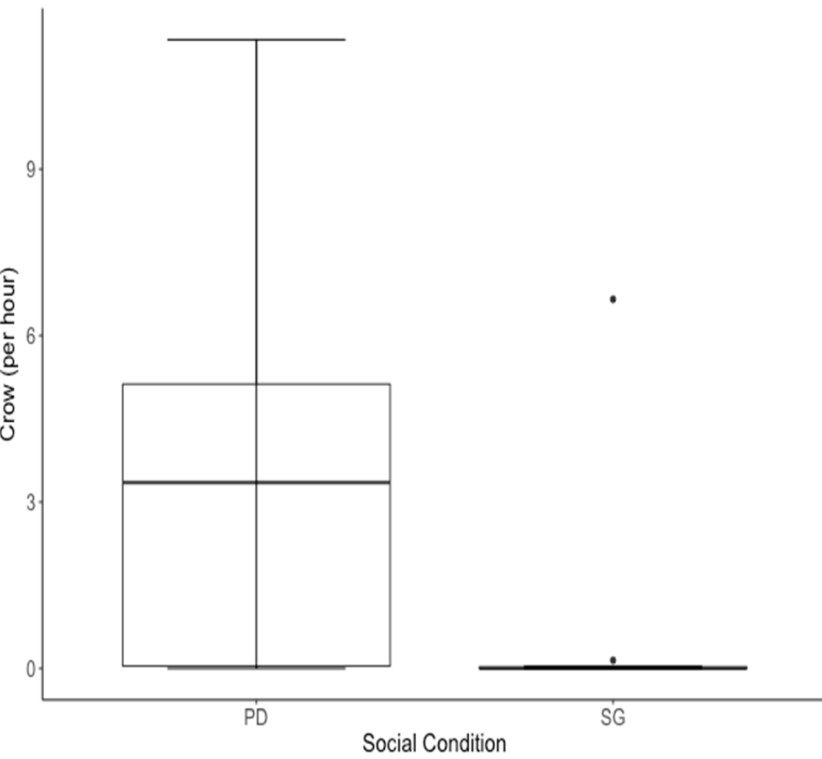

**Figure 5  Median rate of crows per hour for 10 males of cohort I was higher when males were housed in the Perceived Dominance (PD) condition then when housed in the Social Group (SG) condition.** The black horizontal line represents the median value. Whiskers represent the lower and upper quartiles (25% and 75%).

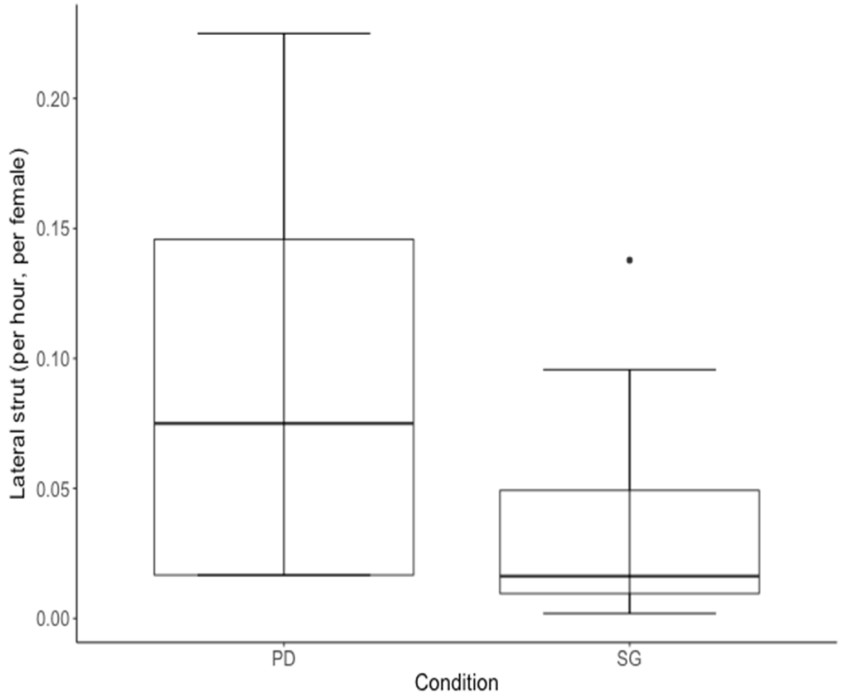

**Figure 6** **Median rate of lateral struts performed per hour (adjusted for female density) for 10 males of cohort I was higher in the Perceived Dominance (PD) condition compared to the Social Group (SG) condition.** The black horizontal line represents the median value and whiskers represent the lower and upper quartiles (25% and 75%).

cannot conclude that adult social rank is a consequence of a male's spatial learning performance. However, when we manipulated social rank so that all males experienced conditions synonymous with high dominance rank (uncontested territory and access to females), we found that a male's learning performance continued to be predicted by their captive social rank, even when this social rank was recorded after cognitive testing. Additionally, the number of females that single males were housed with during the rank manipulation also predicted cognitive performances, with males accompanied by larger groups of females exhibiting higher probabilities of making correct choices. Our findings do not allow us to decipher cause and consequence of the relationship but suggest that spatial learning performances and social rank may become associated over time and the relationship persists even when the direct social pressures associated with social rank have been removed.

The prior attributes hypothesis describes that a high correlation coefficient between an attribute and social rank, is indicative of that attribute having assisted in rank formation (*Chase, 1974*). This hypothesis originally focused on morphological attributes that were developed prior to the formation of the hierarchy but were measured while the dominance hierarchy is established and active, with the inference that these traits assisted in the establishment of the dominance hierarchy. Cognitive performances have been suggested to determine social success (*Byrne & Whiten, 1988*; *Call, 2001*; *Cheney, Seyfarth & Smuts,*

**Table 5** Results from full and minimum adequate model of a generalized linear mixed model Results from full and minimum adequate model of a generalized linear mixed model fitted on the effects of inferred social rank, cohort, the number of females housed with and trial number on binary spatial discrimination task performances for adult male pheasants tested while in the perceived dominance (PD) social condition. Random intercepts and fixed slopes model. Trial and mean Elo-rating were standardised (*z*-scores). The table shows model estimates and standard (SE) for each variable with odds ratio (OR) with low (Lo CI) and high (Hi CI) confidence intervals.

| Predictor variable | Estimate | SE | OR | Lo CI | Hi CI |
| --- | --- | --- | --- | --- | --- |
| **Full model** | | | | | |
| Cohort | −0.219 | 0.294 | 0.803 | −0.914 | 0.476 |
| Correct on first trial | 0.208 | 0.187 | 1.231 | −0.234 | 0.650 |
| **Minimum Adequate model** | | | | | |
| *Intercept* | 0.881 | 0.131 | | | |
| Trial | 0.844 | 0.101 | 2.326 | 0.605 | 1.083 |
| Mean Elo-rating | −0.144 | 0.108 | 0.866 | −0.399 | 0.111 |
| Female (4 females) | 1.002 | 0.216 | 2.723 | 0.491 | 1.513 |
| Trial * R Elo-rating | 0.318 | 0.094 | 1.374 | 0.096 | 0.540 |

*1986*; *Dunbar, 1998*; *Humphrey, 1976*; *Seyfarth & Cheney, 2002*; *Taborsky & Oliveira, 2012*). However, testing whether cognitive performances can predict social rank at the individual level is difficult, and to date, this has not been shown (*Chichinadze et al., 2014*). For cognitive or behavioural attributes, which are highly plastic, measures must be collected prior to the formation of the hierarchy to avoid the confounding possibility that the expression of the attribute is simply a result of social rank. Dominant individuals outperform subordinate individuals on spatial learning tasks (*Barnard & Luo, 2002*; *Fitchett et al., 2005*; *Francia et al., 2006*; *Langley et al., 2018a*; *Spritzer, Meikle & Solomon, 2004*). If the ability to discriminate between spatial cues early in life is beneficial to determining social rank in pheasants, then we expected that cognitive performance of chicks would predict adult cognitive performance and their future social rank. Generally, our findings do not support this. Although there was consistency between chick and adult cognitive performances in their task accuracy, thus suggesting that this ability is fixed across an individual's lifespan and could determine social rank, individuals' 'speed' to reach a learning criterion was not consistent. Furthermore, chick cognitive performances did not predict their estimated social rank as adults. We emphasise that our sample size was small and the results should be interpreted with caution, but if this is a general pattern we suggest two interpretations of these findings.

Spatial learning ability may be subject to cognitive development (*Nowicki, Searcy & Peters, 2002*) and influenced by experience (*Rowe & Healy, 2014*) in the wild. Thus, learning performances assayed during early life while individuals were housed in controlled and identical conditions may not be representative of learning performances that influence a male's ability to attain/maintain a particular social rank as an adult, i.e., spatial learning performances may predict social rank but are not necessarily consistent across an individual's life. Further testing of spatial learning abilities at various life stages and at which point they predict social rank are required to better understand this relationship.
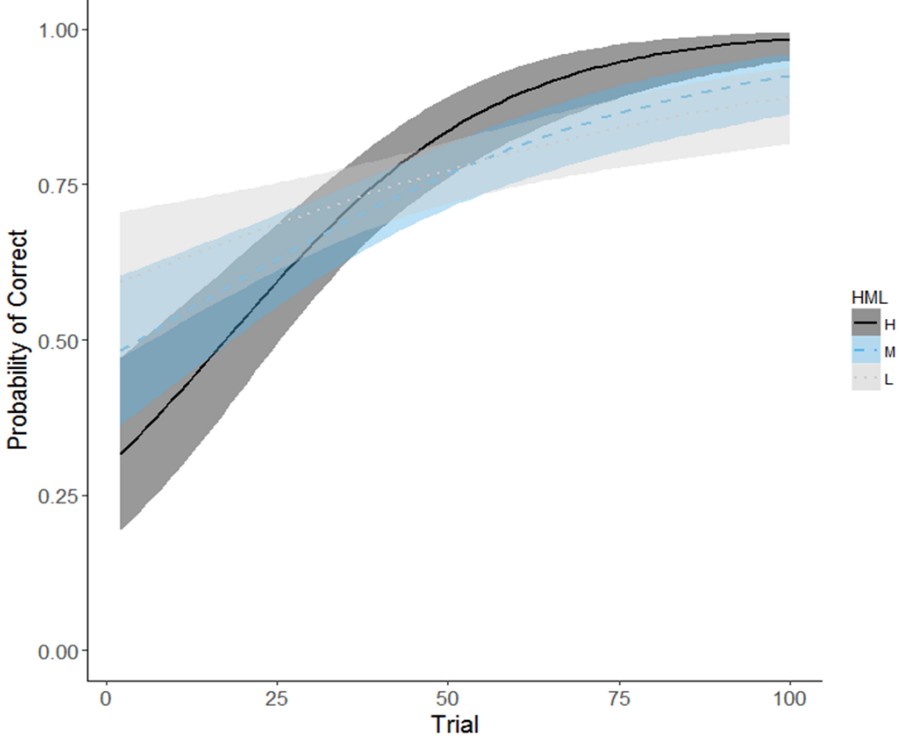

**Figure 7** **Predicted probability of choosing correctly on binary spatial discrimination task for adult male pheasants.** Curves predicted from a generalised linear model with social rank included as a factor with three levels, visualized in ggplot2 (*Wickham, 2009*). Boundaries for high, middle or low rank determined by splitting full range of mean Elo-ratings in to thirds for each cohort. Solid, dashed and dotted lines represent high (Cohort II: $n = 2$), middle (Cohort I: $n = 1$; Cohort II: $n = 2$) and lowest (Cohort I: $n = 2$; Cohort II: $n = 1$) ranking males, respectively.

**Table 6** **Group social rank for adult male pheasants of both cohorts, the number of females they were randomly assigned during the perceived dominance condition (PD) and first choice on a binary spatial discrimination task group social rank for adult male pheasants of both cohorts, the number of females they were randomly assigned (2 or 4) during the perceived dominance condition (PD) and first choice on a binary spatial discrimination task (prior to the opportunity for learning).** Males of cohort I were unknown birds that experienced the social group condition (SG) before cognitive testing in the perceived dominance condition (PD). Males of cohort II are known males that experienced these conditions in reverse.

| Cohort | Rank | Females | First choice |
|---|---|---|---|
| II | 2 | 4 | 0 |
| II | 3 | 4 | 1 |
| II | 4 | 2 | 0 |
| II | 5 | 2 | 1 |
| II | 6 | 4 | 0 |
| I | 7 | 4 | 1 |
| I | 10 | 2 | 0 |
| I | 12 | 2 | 1 |

Alternatively, the ability to discriminate between two locations in a food-motivated task in early life capture realistic processes and skills that are not influential in attaining a high social rank. Although high levels of aggression are associated with superior spatial learning performances in adult animals (pheasants, *Langley et al., 2018a*; *Langley et al., 2018b*; mice, *Francia et al., 2006*; meadow voles, *Spritzer, Meikle & Solomon, 2004*; mountain chickadees, *Pravosudov, Mendoza & Clayton, 2003*), spatial learning performances early in life may not correspond to future social interactions involved with hierarchy formation. If learning performances do influence social rank but learning in the spatial domain is unrelated to learning performances in other domains (domain-specific cognition, *Shettleworth, 2010*), then performances during early life on tasks assaying other cognitive domains may better predict an individual's future social rank. For example, social learning abilities which inform individuals' partner choices and the outcome of social interactions (fighting fish, *Oliveira, McGregor & Latruffe, 1998*) may better predict social rank. The lack of a relationship between early life spatial learning performances and adult social rank may suggest that the relationship we observe in the adults is driven by the influence of aggression on spatial learning performances, rather than the reverse.

We also attempted to address whether the relationship between spatial learning performance and social rank in adults is as a result of current social rank. Our findings offer some support that the current social environment causes individual variation in cognitive performances. Inter-individual variation in learning performance was affected by the number of females a male was housed with during cognitive testing. Males housed with four females had a higher probability of choosing correctly than males housed with only two females. The mechanisms behind this effect are unknown (see *Langley et al., 2018b*), but this suggests that the current social environment has direct influences on individual variation in cognitive performances.

Further indication that the social environment influences variation in cognitive performances, comes from the comparison between this study and *Langley et al. (2018a)*. *Langley et al. (2018a)* found that higher ranking males demonstrated greater learning accuracy but did not learn at a faster rate than lower ranking males. In the current study, more dominant males learned spatial discriminations at a faster rate than lower ranking males; shown by the significant interaction between trial number and social rank. The subtle differences between the two studies may be due to the differences in the social conditions experienced while cognitive performance was assayed. *Langley et al. (2018a)* assayed cognitive performance while males were living in a social hierarchy and under direct pressures of maintaining and acquiring resources. In this study, cognitive performance was assayed while males were not experiencing direct social pressure from other males. This suggests that the rate of learning differs between males of different social rank only when males are tested away from the direct pressures of the social hierarchy. Alternatively, such differences may be due to the different tasks used. *Langley et al. (2018a)* investigated learning performance in an escape-type task, as opposed to the food motivated learning task used in this study. Investigating the relationship between learning performances and social rank using different cognitive tasks will be useful in elucidating whether social interactions are more strongly related to certain cognitive performances, more so than others. Finally,

differences between the two studies may have arisen because of a difference in the number of trials that were conducted; here we conducted 100 trials which gives us much stronger statistical power when determining the rate of learning. To further understand the effects of social rank on spatial learning performances, repeatedly testing individuals while experiencing different social environments (social hierarchy and manipulated social rank), on cognitive tasks targeting the same cognitive domain with the same task affordances, may be one such approach.

In previous studies supporting the social-state dependent hypothesis, acquisition of social rank (mice, *Barnard & Luo, 2002*) or a natural rank change (crab-eating macaques, *Bunnell, Gore & Perkins, 1980*), influenced changes in individual learning performances. In these cases, social rank was not manipulated. Our finding that variation in performance on a learning task corresponds to rank even following a rank manipulation, indicates that the relationship between social rank and cognitive performance is not necessarily driven by current stressors of living in a social hierarchy. We suggest three explanations for these findings in adults.

First, the effects of social rank on cognitive performances may persist even when the immediate social setting has changed. The effects of social defeat/success on performance on a spatial learning task in mice were found to persist up to 13 weeks after social pressures had been removed and mice were housed individually (*Fitchett et al., 2005*). Pheasants agonistic and submissive interactions in the wild may have contributed to variation in cognitive performances. If carry-over effects of past social interactions are driving inter-individual differences in learning, the time it takes for these differences in learning performance to diminish may also give an indication of how long the effects of social defeat, or success, persist (*Laskowski, Wolf & Bierbach, 2016*; *Hsu, Earley & Wolf, 2006*). Further, the relationship between cognitive performances and social rank was observed regardless of whether social ranking was scored before cognitive testing or afterwards. This suggests that individuals are consistent in their social rank experienced in the wild and in captivity.

Second, the relationship between social rank and spatial learning may be mediated by a third variable that we did not measure or modify. For example, individuals suffering from high parasite load may not only have their cognitive performance affected (bumblebees, *Bombus impatiens, Gegear, Otterstatter & Thomson, 2005*), but also be unable to obtain high social rank (red jungle fowl, *Gallus gallus, Zuk et al., 1998*). Future studies may benefit from manipulating parasite load to examine whether these are contributing factors of performances on cognitive tasks and the outcomes of social interactions.

Third, social-rank-related variation in spatial learning performances may have been evident even while individuals were isolated from other males because the manipulation of social rank was not as successful as we believed. Although we observed increases in dominance behaviours (crowing and lateral struts) from singly housed males, the males were still in auditory communication with neighbouring males. Crowing is a behaviour performed by dominant males and may act to indicate territory ownership to conspecifics (*Heinz & Gysel, 1970*; *Ridley & Hill, 1987*). It is possible that male crows communicate dominance status to conspecifics and males were able to assess their relative rank through neighbouring males' crows and so maintain some form of perceived hierarchy even when

housed away from direct social contact. To our knowledge, the specific information that pheasant crows communicate has not been formally tested.

## CONCLUSION

Performance on a spatial discrimination task during early life does not convincingly predict performance on the same task in adulthood, nor does it predict adult social rank in male pheasants, so we conclude that the ability to discriminate between locations may be flexible across an individual's life and does not necessarily provide an advantage in acquiring a high social rank. Instead, when adult, an individual's spatial learning performance does relate to their position in a social hierarchy, and this variation exists even when direct contests with other males are prevented. We also demonstrate that the number of females accompanying a male, affects the spatial learning performance of males. These two results indicate that the social environment, past and current, explains variation in spatial performances. An individual's cognitive performance is unlikely to be fixed from early life, but rather may develop over their lifespan, possibly mediated by their social interactions, and even in mature adults retain some level of plasticity depending on their immediate social conditions. It remains unclear to what extent spatial learning performance and social rank are causally linked.

## ACKNOWLEDGEMENTS

We are grateful to Pip Laker, Rachel Peden and Kenzie Bess for help with husbandry and data collection. We thank Tim Fawcett for providing statistical advice. We thank the editor and three anonymous reviewers for their help in improving the manuscript.

### Funding
This work was funded by an ERC Consolidator Award (616474) to Joah Madden. The funders had no role in study design, data collection and analysis, decision to publish, or preparation of the manuscript.

### Grant Disclosures
The following grant information was disclosed by the authors:
ERC Consolidator Award: 616474.

### Competing Interests
The authors declare there are no competing interests.

### Author Contributions
- Ellis J.G. Langley conceived and designed the experiments, performed the experiments, analyzed the data, contributed reagents/materials/analysis tools, prepared figures and/or tables, authored or reviewed drafts of the paper, approved the final draft.

- Jayden O. van Horik, Mark A. Whiteside and Christine E. Beardsworth performed the experiments, contributed reagents/materials/analysis tools, approved the final draft.
- Joah R. Madden conceived and designed the experiments, contributed reagents/materials/analysis tools, authored or reviewed drafts of the paper, approved the final draft.

## Animal Ethics

The following information was supplied relating to ethical approvals (i.e., approving body and any reference numbers):

All husbandry adhered to the DEFRA Code of Practice (*DEFRA, 2009*). The work was conducted under Home Office licence number PPL 30/3204 to JRM.

## Data Availability

Open Research Exeter (ORE): DOI 10.24378/exe.143.

The raw data is available as Supplemental Data Files.

## Supplemental Information

Supplemental information for this article can be found online at http://dx.doi.org/10.7717/peerj.5738#supplemental-information.

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
