# Peer review of "The relationship between social rank and spatial learning in pheasants, Phasianus colchicus: cause or consequence?"

_PeerJ, doi:10.7717/peerj.5738_

## Round 0.1 · original submission · Major Revisions

Three experts have reviewed your manuscript and while all praised the novelty and importance of your research question, all also had some serious concerns about the experimental design. Indeed, I found it very difficult to evaluate your protocol because of the lack of clarity with which you described your methods (this sentiment echos comments made by all three reviewers).

I think a diagram or table which details the subject IDs, lays out the experimental conditions as well as the order in which subjects were tested, and which subjects experienced which conditions would help enormously. In particular, while you describe the different age cohorts (chicks and 10-month olds) and different social experience conditions your subjects experience (SG and PD), as well as test order (cohort 1 and cohort 2), it was less clear how these were administered methodologically, or later analyzed. For example, if you raised 194 chicks (line 150), how many did you test and why were only 22 re-captured as adults (line 161)? Furthermore, why were the responses of only 9 adults analyzed if 22 were captured (line 283-284)?

I would like to offer you the opportunity to clarify your methods and results, as well as to strengthen your rationale for testing the competing hypotheses you flag in your Introduction as well as the choice of spatial-discrimination task and the particular dominance behaviors you recorded (these are all concerns raised by the reviewers). Please address these concerns, as well as the other outstanding detailed and helpful comments provided by the reviewers.

Reviewer 1 ·

Basic reporting

Throughout the manuscript the authors present some valid points; however they then fail to develop them. The initial points made in the introduction are not really discussed, nor is the significance of the task itself and its relation with social rank. More importantly, there is almost no mention of the rank-related characteristics that determine much of social animals’ behaviours. Since this is an experiment about social rank, this has to be extensively reviewed in the literature. The writing is sufficiently clear, however, some key sections in the manuscript need more flagging and rephrasing, as they are difficult to follow. Figures are relevant to the manuscript and raw data is accessible, although not really clear on what it refers to.

Below I refer to specific sections of the manuscript that need clarification or amending:

1. Lines 54-74: This whole section doesn’t really contribute much to your point that social rank determines cognitive performance. You only give examples (very good ones) of studies that have assessed this relation but you never discuss why so and so rank would be better at cognitive tasks. What are the life-history characteristics of subordinates or dominants that would make them more apt at a cognitive task? If you want to evaluate whether social rank is a cause of cognitive performance this has to be addressed. Additionally, you mention at the start of the paragraph (line 54) that the “social-state hypothesis” can happen through several mechanisms yet you only elaborate on “stress”. Is the unequal distribution of resources (line 74) also a mechanism? It seems to me that this point can also be linked to stress (i.e. nutritional stress). Again, you need to include some information about why ranks differ in their access to resources.

2. Line 59: In both papers (Bunnell & Perkins, 1980; Bunnell et al., 1980) authors refer to animals as crab-eating macaques. If this is not changed, then the Latin name should at least be added to clarify it’s the same species.

3. Line 65: Latin name is missing for the species mentioned here (i.e. crabs).

4. Line 68: What cognitive ability are you referring to here? This has to be explicit.

5. Line 68: Latin name (or the type) is missing from the species mentioned here (i.e. mice).

6. Lines 80-82: Repetition with line 77. I think you mean task participation, rather than opportunities. Animals may have the opportunity to participate but choose not to do so due to a lack of motivation. This will obviously determine the likelihood of learning.

7. Lines 106-107: You have to elaborate a bit more on some of the life-history traits of pheasants. Are they seasonal hatchers? Why do they acquire their territories in October specifically? Does this apply to your study group?

8. Line 114: Wouldn’t having a bigger home range be a better indicator of spatial learning abilities? What about the satellite subordinates? Wouldn’t they need developed spatial abilities to avoid conflict with other males (i.e. avoid dominant’s home ranges)?

9. Lines 117-121: This paragraph should be moved to line 109.

10. Lines 123-124: The sentence is unclear. Please rephrase: “We investigated whether performance on a spatial discrimination task is suggestive of this ability causing male pheasants’ social rank, or whether it is more likely a consequence”

11. Line 125: How long is the breeding season normally?

12. Lines 142-144: The sentence is unclear. Please rephrase: “Third, we tested whether adult males’ cognitive performance assayed while males’ were housed in the perceived dominance condition, was related to their most recent recording of social rank”

13. Lines 265-267: “During most test sessions, we were required to use a temporary mesh partition that stopped the females from approaching the-” Incomplete sentence.

14. Lines 331-333: It is not necessary to explain how you made a figure. I would remove this from the main text and move it to the figure description.

15. Lines 344-347: How does this relate to cognition? You mention nothing of sexual behaviour associated with rank and/or cognitive behaviour. Is this a determinant of fitness based on differences in rank? Table 4 relates to sexual behaviour but it shows the rate of dominance behaviours. Are these behaviours also used in a sexual context? This needs further explanation or to be removed.

16. Lines 457-458: “…cognitive tasks designed to assay performances in the spatial domain may not correspond to interactions with conspecifics or their use of space on a landscape scale” This sentence is unclear. Please rephrase.

17. Line 496: “Stress provides an obvious candidate” Please reword.


STRUCTURE

1. References need to be formatted appropriately.

2. The “Results” section has a different format from what has been used in the first part of the manuscript.


SUPPLEMENTARY MATERIALS

1. Tables 1 & 2: How are these behaviours different from each other? Table 1 mentioned antagonistic outcomes based on winners and losers, but is this independent of rank (i.e. is the winner equally likely to be a subordinate or a dominant)? There’s even a reference to submissive behaviours in Table 1.

2. Tables 4 & 5: There needs to be a bit more description for what each table is showing. For example, what does OR refer to in Table 5?

Experimental design

This study poses some very interesting questions regarding the relation between social rank and cognition. They are well defined and clear from the start. The idea that cognitive abilities determine social rank has been presented before but so far has remained untested. Previous studies (Bunnell & Perkins, 1980), have recommended rank manipulation as a necessary step to answer such questions, which makes this a highly opportune study. Nevertheless, the methodology section needs a lot more detail, as some parts are really ambiguous.

Below I refer to specific sections of the manuscript that need clarification or amending:

1. Line 126: Why were individuals re-captured prior to the breeding season? It’s possible that their behaviour during this season is not normal as there are looking for mates. Wouldn’t this affect their performance?

2. Lines 131-133: “While males were in this perceived dominance condition and experiencing the same social rank, we assayed their performance on the same task that we had presented to the chicks” You mean the social rank was maintained during the social condition or was it the same social rank as before (i.e. previous to re-trapping)? How was rank determined after being re-captured?


3. Line 155: You mention you reared 194 pheasant chicks. Were all of these tested as well? This need to be addressed here or you need to provide a table that has all the information regarding the sample size used throughout the experiment.

4. Line 160: Ad libitum is the correct term as far as I know.

5. Line 166-167: You need to be explicit that from here on, you are referring to those groups as either “known” or “unknown”.

6. Lines 167-171: This needs more flagging. It reads as though there are 3 conditions: (1) Large group aviary with multiple females; (2) Small group aviary with four females; and (3) Small group aviary with two females.

7. Line 169: 60 males and 40 females in this aviary? This number doesn’t match the sample that was re-trapped.

8. Lines 185-187: “If individuals chose incorrectly, indicated by pecking at the crepe paper of the unrewarded well, the wells were removed and a new pair of wells was presented” Did individuals leave the other well untouched if they chose correctly? If not, then this needs to be addressed here or in the discussion.

9. Lines 201-202: At what time was each session carried out? Satiation affects motivation to participate (see Rowe & Healy, 2014). You need to specify here whether animals received any food before each session, as well as the time between sessions and between trials.

10. Lines 203-205: To what session (i.e. morning or afternoon) do these trials correspond to? Do they include trials of the afternoon session as well?

11. Lines 208-209: “Eleven unknown males and one known male that we captured as adults were assigned to the ‘Social Group (SG)’ condition (the large group aviary)” Why was only one know male included here and how was he chosen?

12. Lines 210-213: “Males were assigned to the social conditions in this way because we were initially only interested in known males and so wanted them to be included in the same hierarchy while in the SG condition” Unclear what this sentence means.

13. Lines 213-218: Were the “Cohort I & II” individuals assigned to the two-female or four-female groups of the PD condition randomly? This needs to be clarified.

14. Lines 228-230: “We only included lateral struts which were directed towards females so this was consistent between social conditions (females are present in both social conditions, whereas multiple males were only present in the SG condition)” Why is this? Is male social rank also determined by male-female dyad encounters?

15. Lines 241-243: “Outside of cognitive testing we also collected behavioural observations on dominance behaviours (Table 2), for cohort I males” Again, you need to either elaborate more on the life-history characteristics of pheasants (i.e. differences between sexes). Are males always more dominant than females? This is not clear so it can be interpreted as though females can occupy higher ranks than males.

16. Line 246: “Observations begun at variable times of the day” Aggression levels may change throughout the day. Is aggression consistent despite the time of day? You need to elaborate why this was part of the experimental design.

17. Lines 254-255: “The apparatus was located between two opaque screens so that it could only be approached and viewed by a bird ‘front-on’” Was this the same setup as the one used when they were chicks? If not, then you need to address the differences in design.

18. Lines 256: Why wasn’t the auditory cue used again? This seems to be a completely different setup than the one you tested chicks with.

19. Lines 261-262: “…the males proved difficult to test and appeared distracted by females during the breeding season” It seems to me that this was a possibility from the start. A subordinate’s access to females is male is normally restricted by dominants, so it seems likely that when housed alone with females and out of sight of other males, he will prioritise sexual encounters. Why test individuals at all during this period?

20. Lines 265-267: “The use of this mesh partition did not appear to be stressful as males readily engaged in cognitive testing shortly after the partition was implemented” But not all males participated right? This needs to be clarified, otherwise it contradicts what you just wrote in lines 261-262.

21. Lines 268-269: “Each session consisted of 20 trials. Individuals received one session per day, for five days, resulting in 100 trials in total” Why is this number of trials different from the one before? That’s 70 trials more than when they were tested as chicks. Can you justify this number?

22. Line 295: You need to specify whether the final trial (X) was evaluated in the chick phase, the adult phase or both.

23. Line 299: Same as above but in relation to Y = 80.

24. Lines 310-312: “We fit a generalized linear mixed model (GLMM) with a binomial error structure and a logit link function to assess whether adult learning performance (correct: 1 yes / 0 no) was predicted by group social rank” Were only animals in the SG group used here? This needs to be specified.

25. Lines 322-325: “We included choice on first trial (correct: 1 yes / 0 no) to control for random choice on this first trial; as this trial was prior to the opportunity for learning but may affect subsequent performance on the task and this left the trial variable with trial number 2 – 100 (after trial 1 was removed)” Needs to be moved after the sentence above (line 314).

26. Line 325: Not sure it is necessary to explain that the model failed to converge. Instead, you could write something along the lines of: “To facilitate convergence, we converted trial number, etc.”

27. Lines 333-334: “This model was conducted on eight adult males that each completed 100 trials” These males belonged only to the PD condition?

28. Lines 339-340: “Similarly, we did not include cohort in the model due to the small sample size” I thought the Cohorts were made to increase sample size? Surely the inner state (i.e. stress) of individuals who went from the SG to the PD is not the same as those ones who had the opposite treatment. This could have affected those individuals´ performance and should be controlled for here.

29. Lines 340-342: “Instead, we conducted a Wilcoxon Rank Sum test on the difference between cohort I (n = 3) and cohort II (n = 5) on their overall performance on the task (X=Final)” How did you control for the different groups in the PD condition (i.e. two-female, four-female)?

30. Lines 342-344: “Cohort accounted for the order in which males experienced the social conditions as well as their rearing history (i.e. whether they had experience this task as a chick)” This should be moved to the “Adult social conditions” section.


GENERAL COMMENTS:

There is a lot of inconsistency between the general description of the experiment and statistical analyses over the total sample size analysed at each stage and for each Cohort. A table that describes exactly the number of individuals used in each analysis in both stages is needed.

Validity of the findings

The overall manuscript has methodological issues that need clarification and further explanation. I have a big concern over the lack of consistency throughout the experiment, particularly over the difference in trial number between the chick and adult phases and how individuals were assigned to the cohorts. I believe the authors need to justify why animals were tested so extensively during the adult phase and provide more detail as to how differences in conditions were controlled statistically or in the experiment. I found the discussion rather ambiguous and not going into much detail. There seems to be a lot of ideas presented, but none of them are really developed, nor do they refer back to the literature or the main research questions. The results and methodology are quite interesting and there is sufficient literature for them to be well discussed.

Below I refer to specific sections of the manuscript that need clarification or amending:

1. Lines 382-383: “Of the ten males of cohort I, two males never crowed in either social condition…” What about on the two-female/ four-female conditions?

2. Lines 385-388: You repeat here what you’ve stated in the previous sentence. Is this a different result?

3. Lines 392-394: Same as in the previous comment.


4. Lines 422-424: “Our findings suggest that the effects of past, as well as current social interactions, are important in shaping inter-individual variation in performances on spatial cognitive tasks” Since this was not observed in either cohort, it must be the rank before trapping right? Also, you suggest above (line 418) that performance is predicted by the most recent recording of social group rank. Which record would this be? There needs to be a lot more flagging here and throughout your results section regarding the cohorts and past ranks. At the moment it is difficult to understand what you are referring to.

5. Line 426: “Cognitive performances have been suggested to determine social success“ Do you mean having a higher dominance rank? If so, then wouldn’t this suggest that cognitive performance is a determinant of social rank?

6. Lines 432-435: “This hypothesis originally focused on morphological attributes that were developed prior to the formation of the hierarchy, but were measured while the dominance hierarchy is established and active, with the inference that these traits assisted in the establishment of the dominance hierarchy” This should be moved to the start of the paragraph and the hypothesis mentioned here has to be specified.

7. Lines 436-438: “…measures must be collected prior to the formation of the hierarchy to avoid the confounding possibility that the expression of the attribute is simply a result of the hierarchy” This varies according to the species and in some cases, physiological differences (i.e. sex), so this suggestion is a bit unrealistic. If you want to argue this point then you must address species-specific differences here and what this means for cognitive abilities.

8. Lines 439-440: “…we expected that cognitive performance of chicks reared in controlled conditions, would predict their future social rank” You mean because animals were all released with a similar physical state (all of them received the same amount of food during this stage)? How can you then be sure that the hierarchy is not affected once other factors are introduced (i.e. food availability)? You would have to assume that chicks who were successful at spatial learning (based on the acquisition of territories) were equally successful at foraging challenges. This needs to be addressed when you propose this hypothesis.

9. Line 445: “…small sample sizes and some extreme outliers may have confused the relationship” Not sure what you mean by “confused”.

10. Lines 445-447: “The number of trials it took for individuals to reach a learning criterion (Y=80) showed a strong positive relationship” How do you justify so many trials in the adult phase? Even for a captive study, 100 trials per individual seems excessive. Is it really indicative of spatial learning abilities? The inconsistency between both phases makes it look as though they are separate studies. Additionally, if you only analysed the first 20 trials in the chick phase, why didn’t you analyse the same trials in the adult phase as well? If adults learned the task past trial 20 (or even 30) you would naturally fail to get a relation between both phases. I would suggest running this analysis again with an equal number of trials in both phases.

11. Lines 447-451: This paragraph should be moved to the results section, although I would be more inclined to remove altogether as it offers nothing to your discussion.


12. Lines 459-460: “Perhaps performances on tasks assaying different cognitive domains, such as the ability to discriminate between social stimuli would better predict an individual’s future social rank” This is assuming social rank is not already established from an early age (i.e. when it is inherited). So far you have not mentioned why discrimination of social cues would be relevant for gregarious animals with hierarchies (i.e. avoid aggression, foraging)

13. Lines 465-467: “To truly examine whether cognitive performance predicts social rank, performance on a cognitive task needs to be assayed before any exposure to conspecifics” This is a bit unrealistic. Even in captivity gregarious animals are kept in social groups to avoid stress. You would have to test animals that have been in social isolation since birth. That’s also assuming you have species in which maternal-rank is not inherited. Since you just suggested that discrimination of social cues would have been a better test this seems out of place.

14. Lines 470-471: “…spatial learning ability is flexible across an individual’s lifespan and indeed is influenced by social interactions” But your animals were tested in isolation no? You haven’t analysed whether the interactions they have before each test affected their performance.


15. Lines 496-497: “individuals suffering from nutritional stress or high parasite load not only have their cognitive performance affected…” But you provided food ad libitum here. Presumably, individuals were not nutritionally stressed when you tested them. Were they weighted when they were recaptured from the wild and also when they were moved from the SG to the PD condition to determine this? Also, you don’t make any relation to social rank here. Are subordinates suffering from nutritional stress or have a higher parasite load? Stress resulting from aggressive interactions seems like a more appropriate reference here.

16. Lines 508-509: “…suggests that individuals may not remember this task from early life” This is a completely different explanation from what you have been arguing so far and you don’t elaborate any further than that.

17. Line 509: To what relationship are you referring to here? You mean the relation between performance and cohort?

18. Lines 525-527: “It is possible that male crows communicate dominance ability to conspecifics and males were able to assess their relative rank through neighbouring males’ crows…” It seems to me that auditory cues are a relevant part of establishing dominance. A discussion on how auditory cues may be used for spatial learning by dominants would be relevant here. You used an auditory cue during the chick phase testing but then you switched to a visual cue in the adult phase testing. Why was this, and could it have affected participation? On a different note, dominance is not an ability. A better word would be “status”.

19. Lines 534-537: “Perhaps the reduction in male-male social pressure takes longer to impact cognitive performances, compared with the effect that increased access to females has on cognitive performances, indicating that inter-sexual interactions are a stronger predictor of male cognitive performance than intra-sexual contests” You cite no references to support this observation. Why would an increase in the access to females impact cognitive performance? You never mention anything about the relation between cognitive abilities and reproductive success (i.e. Cauchard et al., 2013). You also never test the effect of inter-sexual interaction on cognitive performance so you cannot confirm this. If this is part of another study, I would remove it from here.

Additional comments

I think this study is relevant to the field of cognition and offers a novel perspective on the relation between rank and cognition. Testing animals under any condition is always challenge and in this case, you not only trained and tested individuals since early age, but later recaptured them and tested each one over an impressive 100 trials! That, to me, seems like a major success on its own. I do feel however that the manuscript lacks a lot of structure and detail. The methodology section in particular is difficult to follow and that sets the tone for the results and discussion section. While many of the comments I have are limited solely to clarity, I still suggest some major revisions to the manuscript are needed.

Reviewer 2 ·

Basic reporting

This is a nicely written study on the causal link between social rank and cognition in pheasants. I feel the authors nicely work towards their research question and propose a nice way of studying specifically the difference between cause and consequence in the relationship between social rank and cognition. I do, however, have some serious reservations about the study, especially regarding experimental design and the validity of the data (see below).

With regard to the introduction:
What I do not really understand is why they specifically choose a spatial learning task, as there is no real reason to assume a link between spatial memory/learning and social rank. Instead, I would have liked a task battery that gives us a more general idea of variation in cognition (which they did with the chicks), or if only one task a task that related more to social cognition (e.g. transitive inference). The authors seem aware of that themselves as they discuss this in l. 442, which raises the question why they nonetheless decide to do so. They mention that in a previous study (of themselves) they already did find a relationship between a spatial task and social rank, yet that in this study they could not differentiate cause from consequence. Given that though, I'm surprised that in this study they use a different experimental task (see l. 468).

The methods section lacks clarity: e.g. I think a picture/sketch of the apparatus/task would greatly benefit the readers' understanding of that task. Furthermore, the multitude of manipulations (2 or 4 females), conditions (SG vs. PD), backgrounds (tested as chick or not), and cohorts makes it very confusing, especially since the sample sizes across all this are not consistent. I suggest including a table in which, per individual everything is set out, with totals that then clearly show the sample sizes per performed tests.

Experimental design

The experimental design is very messy, which seems partly due to the fact that experimental designs of different studies are somehow intertwined. In fact, I would suggest publishing these two? studies as one.

For the current study, I, for example, don't see why animals are placed in differently sized groups in the PD condition, nor why they are placed in different cohorts. And if you choose to do the latter, the animals should have been placed in either cohort in a counterbalanced way with regard to whether they were tested as a chick or not. Consequently, ll. 203 - 206 make no sense to me.

ll. 217-223 are very confusing: What is the reason for taking these agonistic data towards females and not those towards males. Only later in the ms this becomes clear, but this needs to be explained already here, and a clear distinction should be made between data for calculating dominance rank and data for agonistic/self-aggrandizing behaviours.

Why were the animals from the different cohorts housed for different time periods in the PD condition before testing, and is this additional variable included in the analyses

Validity of the findings

I feel that the validity of the findings is harmed by the messy design, as it has created samples for some of the tests are really small. For example, the lack of consistency in spatial learning abilities between chick and adult is based on a spearman's correlation (where an Intra Class Correlation (ICC) would have been better) on only 6! individuals. Whereas the authors do mention this in their discussion, they nonetheless make strong conclusions based on this result (see conclusions and abstract), which they should tone down heavily.

Similarly, due to including the different group sizes in the PD condition, they increase the number of variables and thus decrease the power of their results, especially given the low sample size. The authors acknowledge this themselves and therefore do not include this variable in their GLMM's. Yet, subsequently they show that this manipulation has an effect on spatial learning and thus is a confound of the data used in the GLMM's. Consequently, the results of these GLMM's are not so easily interpreted; for example the interaction effect they find may be due to group size, if the higher ranking individuals have been randomly assigned to the larger group sizes. Again a table with all individuals, their rank, and the different manipulations may help us out here. Nevertheless, as it is a significant confound it should be included in the model.

Overall, given the debatable validity of their results the authors should tone down their conclusion and should especially be careful with generalizations; e.g. why doesn't the title mention that this is a study on pheasants, yet instead presents this as an overall study.

Additional comments

Some minor comments:

please provide latin names of species in intro.

l. 123-124. How well do the dominance ranks calculated form captivity represent what is happening / happened prior to that in the wild?

Methods section: it is ad libitum and not ad libertum

l. 255 .....approaching the APPARATUS while.....

l. 428 given that we don't know what these animals' social rank in the wild was, I would prefer it if you mention ESTIMATED rank here.

Reviewer 3 ·

Basic reporting

The data files provided appear to be complete, although there is no description of variable names (ie. there is no READMe file) to assist in interpreting their data.

There is some awkwardness in the language, such as in lines 49-50 (“the stress experienced”), and line 184. The manuscript is generally very well written, although there are several mis-placed commas (eg. lines 22, 223, 384, 398).

Please provide the latin name for all species listed.

I found some sections to be quite confusing. For example, line 56 describes “several mechanisms", but it is not clear what those mechanisms are in the following sentences. In lines 80-82, an example of this statement would be helpful for the reader.

There is excellent justification in the Introduction for the Social state-dependent hypothesis (lines 54-82) but very limited justification for the prior attributes hypothesis. Please expand on this area (lines 84-94). I would also suggest expanding the first paragraph to better place the study in the field; the current opening paragraph is brief and provides limited context. Why would you expect spatial cognition, specifically, to be linked to dominance rank?

Please provide a citation at the end of line 118.

There are no citations in reference to the cognitive test. Is there any validity that this is actually a test for spatial cognition? Have others used an apparatus like the one used in this study? A figure of the apparatus would also be helpful. It is unclear why there are parentheses around “top” and “bottom” in lines 173 and 174.

The description of the adult social conditions (lines 201-210) is very confusing, when all that needs to be described is that adult males were placed in one of two cohorts, which experienced the social group treatments in reverse order.

The behaviours described in the ethogram in Table 1 actually describes two behaviours (eg. “threat/lunge” describes both “threat/lunge” and “flee/avoid”). However, threat or lunging behaviour can occur independently of fleeing or avoiding.

Experimental design

I am concerned as to why lateral struts exhibited in the PD condition were considered a dominance behaviour. Lateral struts can be used in multiple contexts to convey different signals. For example, when directed toward males, it may be a display indicating quality and thus strength (and potential to win in a fight), whereas it would display quality as a mate when directed toward females. Please address this concern with citations from the literature. Is this different from sexual behaviours mentioned in the methods (line 333)?

While many aspects of the statistical analyses are quite thoroughly described, it is unclear how the data specifically were incorporated into quantifying social rank. Were all behaviours described in Tables 1 and 2 included? Also, please elaborate on the equations described in lines 286 and 289. What do b0 and b1 represent?

Were observations in the PD condition randomized or was there any pattern to the observation periods (line 237)?

Please provide additional information on where/how the pheasant chicks were obtained (line 150). If they were hatched in the wild or in captivity could lead to different cognitive development and abilities.

How did you know the adults caught were 10 months of age, and not 22 months of age (line 160)?

Cohort 2 experienced the PD condition for twice as long as cohort 1. Given the effects of the PD condition, particularly on the results to the cognitive test, was there any effect of this difference? Please address.

Validity of the findings

no comment.

Additional comments

In this study, the authors assess the relationship between performance in a cognitive task in chicks and adults and their social rank in male pheasants. This study is well placed in the literature and appears to be the only study comparing two previously established hypotheses (the ‘Social state-dependent hypothesis’ and the ‘prior attributes hypothesis’). The authors do an excellent job of describing the hypotheses, how their research question and methods relate to each of the hypotheses and predicted outcomes, and where their results sit in the current literature. Although they had a limited sample size (n= 6, 8 and 8 for each of the research questions), they acknowledge this limitation and speculate appropriately in the discussion. As aggression is a highly plastic trait, the have provided new insight into how the social environment shapes cognitive performance.

---

## Round 0.2 · Minor Revisions

Two reviewers who reviewed your original submission have now reviewed your revised article. Both were impressed by the changes that you have made to your article in terms of improving the clarity of your report, and I agree with their determination. Both reviewers, however, still have a few outstanding questions and comments that I ask to you respond to at this stage.

Reviewer 1 ·

Basic reporting

Literature references still need very minor formatting, as do two of the figures presented. Otherwise, I find the general structure and reporting of the manuscript well presented.

Below are the relevant comments for each corresponding section on the tracked manuscript:

Introduction:
1.L64-67: This is difficult to follow. In the first sentence (L64) you write that social rank is a consequence of cognitive abilities; while in the second one (L66) you say it is because of differences in dominance abilities. Can this be all incorporated into one idea? For instance, dominance abilities require certain cognitive abilities as you suggest below.
2.L147: “Motivation” is wrongly used here. In the example you cite, (Drea & Wallen, 1999) subordinate animals had a poor task performance when tested in a group because the presence of a dominant increased the chances of aggression, so they “played dumb” likely to avoid aggression not necessarily because they were unmotivated. Perhaps change motivation to something along the lines of social risk?
3.L215: Missing a word. It should read: “The ability to discriminate….”
4.L279: Typo in the word “conditions”.

References:
1. References still need a bit of formatting. Some of them provide the DOI; others have the phrase “Retrieved from”; and others have neither. In L891, the journal name is abbreviated. Is this correct?

Figures:
1.Figure 1. This figure nicely illustrates how you conducted each experiment in each state. Two points: (1) The caption should include a reference for the opaque colouring of males in the PD conditions; (2) in the manuscript you mention that you were able to reliably test 6 known males (L400) of cohort II, but in the figure, you have only 5 males. Is this correct?
2.Figure 3. Shows only a blue line.

Experimental design

No comments

Validity of the findings

I believe the authors followed the comments and suggestions from both reviewers satisfactorily and now present a much more robust statistical analysis and results. There are some points that remain unclear, albeit, these are relatively minor comments that should not change the overall results. The new discussion is a lot more developed than before and is linked much more nicely to the original questions and the corresponding results.

Below are the comments for each corresponding section on the tracked manuscript:

Methods:
1.L327: Were differences in the testing time controlled for in the analysis? Participation may have been determined by satiation (i.e. they were hungrier in the morning tests compared to the afternoon ones). If they were equally likely to participate regardless of the testing time then perhaps a couple of lines (or numeric data) mentioning this would be good.
2.L330: I think you have to be very explicit here that spatial learning was only assessed in the PD condition because, at this point, it is still unclear whether you test animals in both SG and PD. Perhaps a sentence after you present both conditions could incorporate this (e.g. L384-385).
3.L453-454: I understand the difficulties in testing the same animals repeatedly both, as chicks and adults, and believe you have successfully justified your sample size; however, I am concerned you technically have a sample size of 1 individual in cohort I for the analysis that compares X & Y with the adult mean Elo-rating. Can you be sure the differences in cohort did not affect the social rank acquisition?

Discussion & Conclusions
1. I wonder if there is much point in reporting the results associated with the number of females in the PD condition if they are part of another study. You repeatedly mention a significant relationship between the number of females in this condition and task performance, but what is then its relation to your initial questions regarding social rank? At the moment it is not very clear to me where this fits in the general findings and discussion of the manuscript.
2.L485. Very minor comment. Since your conclusion is basically that chick performance does not determine social rank or adult task performance, I think you have to be clear that you are talking about a relatively recent past social environment. Otherwise, readers might interpret this as a carryover effect from a young age, rather than differences experienced as adults.

Additional comments

I very much enjoyed reading this new manuscript. I think you incorporated all of the comments nicely and made the whole text a lot clearer and concise. Having those extra details really made the experiment and the initial ideas stand out.

Reviewer 2 ·

Basic reporting

n.a.

Experimental design

n.a.

Validity of the findings

n.a.

Additional comments

I Think the authors have done a nice job revising their manuscript and dealing with my comments. I have only some minor comments left:

l. 128 & 1126. Please say long-tailed macaque

l. 182. I don’t see how this is relevant for this study; i.e. studying the causal link between rank and cognition. The subordinates clearly posses the ability, yet inhibit it when a dominant is present to avoid competition over the food to be gained from the task.

l. 450 ff. why did the author’s not counterbalance the place of reward over the birds? And did the authors control for olfactory cues?

l. 638. you just said only 8 adult individuals participated reliably, why then now suddenly an n of 9. Similarly, you mentioned before that only 7 of the known individual’s completed all trials as a chick and that looking at the first 20 trials only increased your sample with 1, leading again to a sample size of 8 and not 9. See also line 656

l. 652. This does not correspond to fig 1. where it says that only 5 of the known individuals cognitive performance was essayed.

l. 868. This sentence needs to be reformulated making clear that the manipulation of social rank is a control for the set-up, as to avoid immediate effects of rank within a group.

---

## Round 0.3 · accepted · Accept

It is my pleasure to accept your article for publication in PeerJ. I have reviewed your responses to the two reviewers' comments and believe that you have addressed all thoroughly.

#